# Telomere as a Therapeutic Target in Dedifferentiated Liposarcoma

**DOI:** 10.3390/cancers14112624

**Published:** 2022-05-25

**Authors:** Irene Alessandrini, Stefano Percio, Eisa Naghshineh, Valentina Zuco, Silvia Stacchiotti, Alessandro Gronchi, Sandro Pasquali, Nadia Zaffaroni, Marco Folini

**Affiliations:** 1Molecular Pharmacology Unit, Department of Applied Research and Technological Development, Fondazione IRCCS Istituto Nazionale dei Tumori di Milano, Via G.A. Amadeo 42, 20133 Milan, Italy; irene.alessandrini2@istitutotumori.mi.it (I.A.); stefano.percio@istitutotumori.mi.it (S.P.); eisa.naghshineh@istitutotumori.mi.it (E.N.); valentina.zuco@istitutotumori.mi.it (V.Z.); sandro.pasquali@istitutotumori.mi.it (S.P.); 2Adult Mesenchymal Tumor and Rare Cancer Unit, Department of Cancer Medicine, Fondazione IRCCS Istituto Nazionale dei Tumori di Milano, Via Venezian 1, 20133 Milan, Italy; silvia.stacchiotti@istitutotumori.mi.it; 3Sarcoma Service, Department of Surgery, Fondazione IRCCS Istituto Nazionale dei Tumori di Milano, Via Venezian 1, 20133 Milan, Italy; alessandro.gronchi@istitutotumori.mi.it

**Keywords:** apoptosis, autophagy, dedifferentiated liposarcoma, G-quadruplex, micronuclei, soft tissue sarcoma, telomeres

## Abstract

**Simple Summary:**

DDLPS is a non-lipogenic tumor with aggressive clinical behavior. DDLPS patients have limited therapeutic options, especially those with retroperitoneal tumors, and their outcome remains largely unsatisfactory, highlighting the need for novel treatment strategies. Gene expression analysis carried out in clinical samples from primary retroperitoneal DDLPS indicated that these tumors highly express genes involved in telomere maintenance, the cellular process that guarantees immortality to tumor cells. In this study, we evaluated the effect of RHPS4, a molecule able to alter telomere by binding to telomeric structures called G-quadruplexes, in patient-derived DDLPS cell lines. Exposure to RHPS4 induced DNA damage and decreased cell proliferation and migration, thus suggesting telomere as a novel target and G-quadruplex binders as innovative therapeutic agents in DDLPS.

**Abstract:**

Background: Well-differentiated (WD)/dedifferentiated (DD) liposarcoma (LPS) accounts for ~60% of retroperitoneal sarcomas. WDLPS and DDLPS divergently evolve from a common precursor and are both marked by the amplification of the 12q13–q15 region, leading to the abnormal expression of *MDM2*, *CDK4*, and *HMGA2* genes. DDLPS is a non-lipogenic disease associated with aggressive clinical behavior. Patients have limited therapeutic options, especially for advanced disease, and their outcome remains largely unsatisfactory. This evidence underlines the need for identifying and validating DDLPS-specific actionable targets to design novel biology-driven therapies. Methods: Following gene expression profiling of DDLPS clinical specimens, we observed the up-regulation of “telomere maintenance” (TMM) pathways in paired DD and WD components of DDLPS. Considering the relevance of TMM for LPS onset and progression, the activity of a telomeric G-quadruplex binder (RHPS4) was assessed in DDLPS patient-derived cell lines. Results: Equitoxic concentrations of RHPS4 in DDLPS cells altered telomeric c-circle levels, induced DNA damage, and resulted in the accumulation of γ-H2AX-stained micronuclei. This evidence was paralleled by an RHPS4-mediated reduction of in vitro cell migration and induction of apoptosis/autophagy. Conclusions: Our findings support telomere as an intriguing therapeutic target in DDLPS and suggest G-quadruplex binders as innovative therapeutic agents.

## 1. Introduction

Well-differentiated (WD)/dedifferentiated (DD) liposarcoma (LPS) is one of the most frequent soft-tissue sarcomas (STS) and accounts for approximately 60% of all retroperitoneal sarcomas [1]. WDLPS and DDLPS are thought to divergently evolve from a common precursor and are both marked by the amplification of the 12q13–q15 chromosomal sub-region that sustains the abnormal expression of genes driving tumorigenesis, such as *MDM2*, *CDK4*, and *HMGA2* [2]. In addition, DDLPS harbors further genomic changes, including 6q23 and 1p32 co-amplifications [3]. Recently, the *CTDSP1/2-DNM3OS* fusion genes have been identified in a subset of DDLPS [4]. However, the main genomic events associated with the progression of WDLPS to DDLPS still remain to be clarified. While WDLPS is a low-grade disease resembling mature adipose tissue, which recurs locally, DDLPS is a non-lipogenic disease associated with more aggressive clinical behavior, characterized by multifocal local recurrences and distant metastases [5]. DDLPS often includes both WD and DD tumor components, which can be identified at radiological evaluation and pathological examination after surgery, reinforcing the hypothesis that DDLPS progresses from WDLPS.

The standard treatment for localized DDLPS is surgery, while preoperative radiotherapy has been recently suggested to reduce local recurrence only in less aggressive DDLPS (malignancy grade 2) and in WDLPS [6]. Despite optimal management of primary tumors, roughly half of all patients developed local recurrence and distant metastasis. Patients who are deemed unresectable with surgery are treated with first-line doxorubicin or doxorubicin-based regimens [7,8], which can achieve a tumor response in approximately one in four to five patients [9], while ifosfamide, trabectedin, and eribulin [10] represent further treatment options. In spite of improvements in patient survival over time [11], outcomes remain largely unsatisfactory [6]. This evidence underlines the need for increased knowledge of DDLPS molecular characteristics, which is currently limited, with the final aim of identifying and validating DDLPS-specific actionable targets to inform the design of novel biology-driven therapies.

In the present study, we comparatively analyzed gene expression profiles of paired WD and DD tumor components from primary untreated retroperitoneal WD/DDLPS surgical specimens and found that pathways related to “telomere maintenance” were up-regulated in DD compared to the WD tumor component of DDLPS.

The activation of a telomere maintenance mechanism (TMM) is an almost overriding feature of human cancers [12]. Since normal cells do not usually show sufficient levels of telomere maintenance activity, TMMs have been regarded as intriguing cancer-associated targets [12]. Two TMMs have yet been described in human cancers: Telomerase activity, which is the most common TMM in cancer, and the alternative lengthening of telomere (ALT) mechanism, a homologous recombination (HR)-based process [12]. Telomerase is a multi-protein complex characterized by a reverse transcriptase (TERT) that synthesizes telomeric DNA on chromosome ends using a long non-coding RNA moiety (TERC) as a template [12]. Approximately 90% of tumors show telomerase activity, whereas most of the remaining tumors, especially those of mesenchymal and neuroepithelial origin, rely on the ALT mechanism to maintain their telomeres [12,13]. Cells that maintain telomeres through ALT usually lack telomerase activity and may display a different array of the following molecular features [12,14]: (i) Telomeres with heterogeneous length, which range from very short to more than 50 kilobases; (ii) the presence of a typical subset of promyelocytic leukemia (PML) nuclear bodies, termed ALT-associated PML bodies (APB), which contain telomeric chromatin and telomere- and HR-associated proteins; (iii) a high rate of telomere sister chromatid exchange (T-SCE); and (iv) the occurrence of extrachromosomal telomeric DNA, which may be present in the form of (a) linear double-stranded DNA; (b) high molecular weight “t-complex”; (c) double-stranded telomeric circles (t-circle), and (d) partial single-stranded circular DNA, such as C-circles, which are now emerging as quantifiable markers of ALT status [15,16].

TMMs have been viewed as static properties of cancer, as tumors adopting a single TMM during transformation were believed to maintain it indefinitely [17]. Such a view has been overcome by the evidence that both mechanisms may coexist within a single tumor. Though it remains unclear whether both TMMs are active in a single cell or whether the bulk tumor contains cell populations that use a single TMM [12], this observation implies that the prevalence of a TMM with respect to the other may provide a tumor bulk with distinct, or even opposite, biological behaviors [12]. Moreover, the mechanism underlying the activation of a TMM over the other still remains to be elucidated, even if genetic bases (e.g., TERT promoter mutations, ATRX mutations, telomeric variant repeats) for the activation of either mechanism in cancer are becoming recognized [18].

Since the evidence of selective reactivation in most human tumors, telomerase has gained attention as a possible cancer-associated target, and several strategies to interfere with its expression and functions for potential therapeutic applications have been widely pursued [19]. In this regard, it is worth noting that Imetelstat (GRN163L) was the first telomerase inhibitor to enter clinical trials in patients with solid tumors and myeloproliferative diseases [20]. By contrast, no genuine ALT targeting therapies have been developed yet, due to the fragmentary information available on specific molecular factors involved in the engagement and maintenance of this mechanism in human tumors [12]. Nonetheless, telomeres have been considered biologically relevant targets of small molecules able to interact and stabilize G-quadruplex (G4) structures [21]. The search for telomeric G4-interacting agents has been fostered by the G-richness of telomeric DNA and its propensity to fold into G4 structures [22], as well as by the hypothesis that drug-mediated stabilization of telomeric G4 would allow one to selectively interfere with TMM in human cancer [22].

By taking into account our gene expression data and the documented relevance of TMMs for DDLPS onset and progression [12,23,24], we assessed the activity of a G4 stabilizing agent in MDM2-amplified DDLPS patient-derived cell lines to support telomere as a novel therapeutic target for the disease. Among the numerous compounds tested thus far for their capability to interfere with telomeric G4 structures [25], we focused on the acridine derivative RHPS4 (3,11-difluoro-6,8,13-trimethyl-8H-quino[4,3,2-kl]acridinium methosulfate), which is the most advanced telomeric G4 ligand in terms of preclinical investigations [22,26]. Results are reported herein.

## 2. Materials and Methods

### 2.1. Cell Lines and Chemicals

Patient-derived MDM2-amplified DDLPS cell lines (LS-GD-1 and LS-BZ-1) were established in our lab as previously described [27]. The use of patients’ DDLPS samples to generate cell lines was authorized by the Ethics Committee of the Fondazione IRCCS Istituto Nazionale Tumori (INT), Milan, Italy (Project approval code: INT 131/16). Cells were grown as monolayer cultures in the logarithmic growth phase in DMEM/F-12 medium (Lonza, Treviglio, Italy) supplemented with 10% fetal bovine serum at 37 °C in a humidified 5% CO_2_ incubator. Cells were monitored by microsatellite analysis (AmpFISTR Identifiler PCR amplification kit, Applied Biosystems, Foster City, CA, USA) and checked for the lack of Mycoplasma contamination (MycoAlertTM Mycoplasma Detection Kit, Lonza) periodically.

The G4 ligand 3,11-Difluoro-6,8,13-trimethylquino[4,3,2-kl] acridinium methylsulfate (RHPS4) was purchased as a powder from Selleckchem (Houston, TX, USA). The compound was dissolved in DMSO (Sigma-Aldrich S.r.l., St. Louis, MO, USA) and stored as a stock solution at −80 °C. Working solutions were prepared in a complete cell culture medium just before use.

### 2.2. Gene Expression Profiles 

Paired frozen samples, including the WD component, the DD component, and normal adipose tissue, were obtained from 15 patients with a retroperitoneal, primary, untreated WD/DDLPS who underwent surgery at INT in 2017. Total RNA was extracted from frozen tissues using the RNeasy mini kit (Qiagen, Hilden, Germany). Gene expression profiles were assessed using the GeneChip Clariom S Human Arrays platform. Raw data were normalized according to the RMA algorithm of oligo Bioconductor package [28]. Probes not assigned an official gene symbol were removed, while for probes mapping on the same gene symbol, the one with the highest variance value was selected. The generated dataset was deposited in the GEO database (accession number GSE159659). 

Differential gene expression was estimated both in terms of fold change (FC) and t-value (paired t-test), using the limma Bioconductor package [28]. A threshold of 0.05 was considered for statistical significance using the False Discovery Rate (FDR) correction to adjust multiple testing biases. All analyses were performed using the computing microenvironment R. Over-representation analysis was performed on significantly differentially expressed genes by the Webgestalt online tool [29]; the REACTOME database was investigated, by selecting the total amount of normalized genes as the background and the same FDR value to assess significance. 

A second gene expression profile dataset was retrieved, yet normalized with the RMA algorithm, from the GEO repository (accession number GSE30929), which includes 140 primary LPS with different histologic subtypes, among which there were 52 WDLPS and 40 DDLPS.

### 2.3. Characterization of TMM in DDLPS Cells

Total DNA was obtained from DDLPS cells by Qiagen Blood & Cell Culture DNA Midi Kit (Qiagen, Hilden, Germany), according to the manufacturer’s instructions. Telomere length measurement was performed by telomere restriction fragment (TRF) analysis on 5 μg of genomic DNA digested with *Rsa I*/*Hinf I* restriction enzymes (New England Biolabs, Ipswich, MA, USA) upon pulsed-field gel electrophoresis and Southern blotting, as previously described [30]. Absolute telomere length (aTL) was quantified using the Absolute Human Telomere Length quantification qPCR assay kit (ScienCell^TM^, San Diego, CA, USA), following the manufacturer’s instructions. A human genomic DNA sample of known telomere length (Lot. 23756; ScienCell^TM^) was included as a reference for calculating the aTL of target samples.

The levels of C-circle DNA were assessed by the conventional CC assay (CCA), as described in [15,31]. Briefly, genomic DNA was incubated at 30 °C for 8 h in the presence or absence of 7.5 U of ϕ29 DNA polymerase (New England BioLabs) and heat-inactivated at 65 °C for 20 min. Reaction products were dot-blotted, UV-cross-linked, and then incubated overnight with a ^32^P-(CCCTAA)_3_ probe using the PerfectHyb Plus hybridization buffer (Sigma-Aldrich) under native conditions. After washing, filters were subjected to autoradiography and scanned. Signal intensity was quantified using ImageJ 1.46r. To ensure equal sample loading, filters were then stripped and re-probed with a ^32^P-labelled Alu probe.

Telomerase activity was detected on 1 μg of protein extract following the PCR-based Telomeric Repeat Amplification protocol (TRAP) [30] using the TRAPeze^®^ kit (Merck S.p.a., Vimodrone, Italy), according to the manufacturer’s instructions. PCR amplification products were then resolved on a 10% non-denaturing polyacrylamide gel and visualized by SYBR™ Safe DNA Gel Stain (Invitrogen™, Thermo Fisher Scientific, Monza, Italy) using the Essential V6 Gel Documentation System (UVITEC, Cambridge, UK).

Genomic DNA and proteins from the ALT-positive SK-LU-1 (ATCC^®^−HTB-57™) and telomerase-positive A549 (ATCC^®^—CCL-185™) cells were included as internal controls in each assay.

### 2.4. Assessment of the Expression Levels of Telomerase Components by Real-Time RT-PCR

Total RNA was obtained from DDLPS cells using the Qiagen RNeasy Mini kit (Qiagen) and digested with 20U RNase-free DNase I (Qiagen). Randomly primed total RNA was reverse transcribed using the High-Capacity cDNA Reverse Transcription kit (Applied Biosystems, Carlsbad, CA, USA) and gene expression levels were analyzed by specific TERT (Hs99999022_m1) and TERC (Hs03454202_s1) TaqMan^®^ Gene expression assays (Applied Biosystems). Amplifications were run on the 7900HT Fast Real-Time PCR System (Applied Biosystems). Data were analyzed by SDS 2.2.2 software (Applied Biosystems). RNaseP (TaqMan Rnase P Control Reagents) was used as the normalizer during the analysis of gene expression.

### 2.5. Cell-Based Experiments

To assess the effect of RHPS4 on DDLPS cell growth, 6.0 × 10^4^ cells/well were seeded in 12-wells plates. After 24 h incubation at 37 °C and 5% CO_2_, cells were treated with increasing concentrations (1.25, 2.5, 5.0, and 10 μM) of freshly diluted RHPS4 and incubated for 24, 48, and 72 h. The number of growing cells was subsequently determined by counting the cells in a particle counter (Coulter Counter, Coulter Electronics, Luton, UK), and the IC_50_ (the concentration of the ligand leading to 50% inhibition of cell growth) was calculated from the dose−response curves (% inhibition of cell growth with respect to untreated cells (i.e., cells exposed to DMSO) as a function of the Log10 of RHPS4 concentrations) using GraphPad Prism 5.01 (GraphPad Software Inc., San Diego, CA, USA).

To evaluate the effect of RHPS4 on cell growth over time, LS-GD-1 and LS-BZ-1 cells (7.5 × 10^5^ cell/flask) were seeded in 75-cm^2^ tissue culture flasks and allowed to attach for 24 h at 37 °C, 5% CO_2_. Cells were then treated for 24, 48, and 72 h with an equitoxic amount of RHPS4 (i.e., IC_50_ at 48 h), and the number of growing cells was estimated by a dye exclusion assay (0.4% Trypan Blue Solution, Sigma-Aldrich) using a TC20™ automated cell counter (Bio-Rad Laboratories S.r.l., Segrate, Italy). Untreated cells (i.e., cells exposed to DMSO) were used as controls.

The effects of RHPS4 on the migrating capabilities of DDLPS cells were assessed by wound healing and transwell assays. Specifically, wound healing was performed by seeding 1.0 × 10^6^ cells in a rectangular cell culture plate (Cell Comb™ Scratch Assay, EMD Millipore Corp., Billerica, MA, USA) and monitored by bright-field microscopy. Images of migrating cells in the absence or presence of RHPS4 were acquired immediately after the scratch (T_0_) and 24 h later (T_24_) using the EVOS™ XL Core Imaging System (Thermo Fisher Scientific). Wound healing was quantified by measuring the shortest distance between the edges of the scratch at T_0_ and T_24_ using ImageJ 1.46 r. Data were reported as the % of wound closure in untreated and G4 ligand-treated cells at T_24_ vs. T_0_. For the transwell assay, cells were seeded in the upper chamber (2.0 × 10^5^ cells/well) of 24-well transwell plates (Costar; Corning Incorporated, Corning, NY, USA) equipped with polycarbonate filters, using a serum-free medium. The complete DMEM/F-12 (Lonza) medium was added to the lower chamber of the transwell as a chemoattractant. After 24 h incubation at 37 °C and 5% CO_2_, untreated and RHPS4-treated cells were fixed with 70% cold ethanol and stained with 0.4% sulforhodamine B (Sigma-Aldrich) in 0.1% acetic acid. Images of stained filters were acquired and processed using the EVOS™ XL Core Imaging System. The number of migrating cells was counted at ×40 magnification and reported as the % of migrating cells in treated vs. untreated cells.

### 2.6. Western Immunoblotting

Twenty-five micrograms of total protein extracts prepared according to standard methods were fractioned by SDS-PAGE (4–12% Bis–Tris/SDS polyacrylamide gel, NuPAGE, Thermo Fisher Scientific) and transferred onto Hybond nitrocellulose filters (GE Healthcare Life Sciences, Buckinghamshire, UK). Filters were blocked for 1 h at room temperature in 1× PBS-Tween20, 5% skim milk, and then incubated overnight at 4 °C with primary antibodies: Rabbit polyclonal anti-γ-H2AX (ab11174, Abcam, Cambridge, UK); anti-LAMP-1 (ab24170, Abcam); anti-LC3B (ab51520, Abcam); and anti-PARP-1 (#9542, Cell Signaling Technology, Danvers, MA, USA). Mouse monoclonal anti-GAPDH (G8796, Sigma-Aldrich S.r.l.) and anti-Vinculin (VCL, V9131, Sigma-Aldrich S.r.l.) antibodies were used to ensure equal protein loading. The filters were then probed with secondary, peroxidase-linked whole antibodies (GE Healthcare) for 1 h at room temperature, and blotted proteins were detected using the Novex^®^ ECL HRP Chemiluminescent detection system (Thermo Fisher). Filters were then subjected to autoradiography, the films were scanned, and images were analyzed using ImageJ 1.46r.

### 2.7. Fluorescence Microscopy Analyses

For fluorescence microscopy analyses, 1.0 × 10^5^ cells were seeded on Collagen type I-coated coverslips (Corning, Corning, NY, USA), fixed, and permeabilized with 100% cold methanol. Cells were then probed for 1 h at room temperature with the anti-γ-H2AX (Abcam) antibody or a rabbit polyclonal anti-LC3B (#2775, Cell Signaling Technology) antibody and subsequently with an anti-rabbit AlexaFluor^®^488 (A-11008, Thermo Fisher Scientific) secondary antibody. To assess the telomeric localization of DNA damage, cells were probed with the anti-γ-H2AX (Abcam) antibody and a mouse monoclonal anti-TRF2 (IMG-124A, Novus Biologicals, CO, USA) antibody, washed in PBS, and then probed with anti-rabbit AlexaFluor^®^488 and anti-mouse AlexaFluor^®^594 (A-11005, Thermo Fisher Scientific) antibodies, respectively. Nuclei were counterstained with 0.1 mg/mL of 4′,6-diamidino-2-phenylindole (DAPI, Thermo Fisher Scientific, D1306). Images were acquired by a Leica TCS SP8 X confocal laser scanning microscope (Leica Microsystems GmbH, Mannheim, Germany) in the scan format 1024 × 1024 pixel using an HC PL APO 63×/1.40 CS2 oil immersion objective. Images were analyzed using the software Leica LAS AF rel. 3.3 (Leica Microsystems GmbH, Mannheim, Germany) and processed with ImageJ 1.46r. To evaluate the extent of co-localization (Pearson’s coefficient and Mander’s overlapping coefficient) between γ-H2AX and TRF2, the toolbox JACoP under Image J was used as described in [32].

The occurrence of micronuclei was assessed on DAPI-stained nuclei, whereas apoptotic nuclear morphology was investigated on living cells (4.0 × 10^4^) stained with a propidium iodide solution (50 γg/mL propidium iodide, 1.0 mg/mL RNase, and 0.01% Nonidet P40 in PBS) and spotted onto glass slides. Images were acquired by the Nikon Eclipse E600 fluorescence microscope (Nikon, Tokyo, Japan) using the NIS-Elements software (Nikon). The quantification of micronuclei and apoptotic cells was assessed by two independent observers on 200 cells in at least three different fields (magnification ×100) from three independent experiments. Micronuclei were first scored for DAPI staining and then for the presence of γ-H2AX. Data are reported as the number of micronuclei/100 cells or as the percentage of apoptotic cells within the overall cell population.

### 2.8. Statistical Analysis

If not otherwise specified, data have been reported as mean values ± s.d. from at least three independent experiments. Non-parametric Mann–Whitney and Wilcoxon tests were used to analyze differences between samples. *p*-values < 0.05 were considered statistically significant.

## 3. Results

### 3.1. “Telomere Maintenance” Pathways Are Up-Regulated in DDLPS

We assessed the gene expression profile of a series of 45 samples, including a paired WD component, DD component, and normal adipose tissue (A), from 15 treatment naïve DDLPS using the GeneChip Clariom S Human Arrays platform (GSE159659). Principal component analysis (PCA) showed clear segregation between DD and A samples, while WD samples spanned the intermediate region, suggesting a mixture of characteristics compared to the other tissue types (Figure 1A). Consistently, differential expression analysis showed 23% of genes differentially expressed between DD and A compared to only 1.2% of genes between WD and A. Interestingly, 7% of genes were significantly differentially expressed between WD and DD, indicating an important transcriptional reprogramming in DD components, which may mirror the acquisition of a more aggressive phenotype. 

To provide a holistic characterization and the biological meaning of these differences, we performed an over-representation analysis on the list of up- and down-regulated genes in DD compared to WD tumor components (Figure 1B). Two distinct networks emerged, where nodes represent de-regulated pathways and edges represent the number of shared genes, calculated by the Jaccard index. Consistently with the non-lipogenic nature of DDLPS, an independent sub-network related to adipogenesis and lipid metabolism was found down-regulated in DD components (Figure 1B). Interestingly, important pathways related to “cell cycle”, “DNA repair”, “telomere maintenance”, “epigenetic regulation”, and “WNT signaling” emerged as up-regulated in DD components (Figure 1B; Appendix A). 

Based on the relevance of TMMs for DDLPS progression, and, consequently, prognostic risk stratification [12,23,24], we analyzed the deregulated “telomere maintenance” pathways. In particular, WRAP53 and FEN1 were among the genes related to the maintenance of telomere length and integrity that were found to be up-regulated in DD compared to the WD component. WRAP53, also known as TCAB1, is essential for the activity of the telomerase complex [33], whereas FEN1 is a DNA replication and repair protein known to contribute to telomere stability in both telomerase- and ALT-positive cells [34,35] (Figure 1C). This significantly increased expression of WRAP53 and FEN1 in DD components of DDLPS was confirmed in an independent dataset comparing specific histological subtypes of LPS, which includes 52 WDLPS and 40 DDLPS (GSE30929) (Figure 1C).

### 3.2. Characterization of the Operating TMM in DDLPS Cell Lines

Two patient-derived DDLPS cell models were used to assess telomeres as therapeutic targets for the disease. Firstly, the operating TMM in each cell line was defined by the presence of telomerase activity and/or the occurrence of ALT-associated markers, including TRFs distribution pattern and the presence of telomeric c-circles [12].

The result from the Southern blot analysis of TRFs (Figure 2A) indicated that LS-GD-1 cells were characterized by a TRF length distribution, ranging from approximately 4.3 to >48.5 Kb, typically associated with the ALT phenotype, as illustrated by the TRF analysis of SK-LU-1 lung cancer cells, used as an ALT-positive control [13]. Conversely, LS-BZ-1 cells showed markedly shorter telomeres compared to LS-GD-1 cells, with a distribution of TRFs within a smaller range of molecular sizes (from <2.0 up to 7.0 Kb), though prominently different from the homogeneous TRF distribution observed in A549 cells included as a telomerase-positive control (Figure 2A). These observations were confirmed by the real-time PCR quantification of the absolute telomere length (Figure 2B), showing that LS-GD1 cells have long telomeres (~24 ± 11 Kb), similar to those of SK-LU-1 cells (15.4 ± 1.6 Kb), whereas LS-BZ1 cells were characterized by very short telomeres (~1.5 ± 0.5 Kb), even shorter than those of A549 cells (3.3 ± 0.7 Kb).

To date, the presence of single-stranded circles comprising C-rich telomeric DNA is considered a specific and quantifiable marker of ALT activity [31]. Results obtained by the CCA showed that DDLPS cell lines were characterized by quantifiable levels of c-circle DNA (Figure 2C), though in remarkably lower amounts with respect to those detected in the ALT-positive SK-LU-1 cell line (Figure 2C). Notably, A549 cells included in the assay as telomerase-positive controls showed quantifiable levels of c-circle that were comparable to that of LS-BZ-1 cells.

Moreover, the real-time RT-PCR evaluation of the expression levels of the two main components of telomerase enzyme (i.e., the long non-coding RNA TERC and TERT mRNA encoding for the telomerase reverse transcriptase catalytic subunit) revealed that all tested cell lines do not differ prominently in terms of TERC expression levels (Figure 2D), whereas DDLPS and SK-LU-1 cells showed remarkably lower levels of TERT mRNA compared to the telomerase-positive A549 cells (Figure 2D). Nonetheless, both DDLPS cell lines were characterized by the presence of telomerase enzymatic activity, similarly to the telomerase-positive A549 cells (Figure 2E). As expected, the TRAP assay failed to detect telomerase catalytic activity in the ALT-positive SK-LU-1 cells (Figure 2E).

### 3.3. Biological Effects of Equitoxic Concentrations of RHPS4 in DDLPS Cells

The exposure of DDLPS cells to increasing concentrations (1.25, 2.5, 5, 10 µM) of RHPS4 for 24, 48, and 72 h resulted in a dose-dependent inhibition of cell growth in both cell lines at all considered time points (Figure 3A), with IC_50_ values (i.e., the concentration of the compound leading to 50% inhibition of cell viability) falling within the low micromolar range upon 48 and 72 h of drug exposure (Table 1).

Based on these observations, and to perform a direct comparison of the biological effects, an equitoxic concentration (i.e., the IC_50_ at 48 h) of RHPS4 was selected to perform subsequent analyses in both cell lines.

In order to assess whether the cytotoxic activity of RHPS4 was associated with the induction of DNA damage, the accumulation of γ-H2AX—a specific marker of DNA double-strand breaks [36]—was investigated. 

Western blot analysis revealed that LS-GD-1 cells were characterized by higher levels of basal DNA damage compared to LS-BZ-1 cells (Figure 3B), a feature commonly associated with ALT activity [37]. In addition, the exposure of LS-GD-1 cells to RHPS4 for 72 h did not result in a marked increase in γ-H2AX protein levels (Figure 3B). Conversely, a pronounced increase in γ-H2AX protein amounts was appreciable in LS-BZ-1 cells exposed to an equitoxic concentration of RHPS4 for 72 h (Figure 3B). This evidence was also confirmed by qualitative immunofluorescence analyses showing that the exposure to RHPS4 resulted in a negligible level of DNA damage in treated compared to untreated LS-GD-1 cells (Appendix A), whereas extensive DNA damage was observed in treated compared to untreated LS-BZ-1 cells (Appendix A). Notably, immunofluorescence analysis revealed that the accumulated γ-H2AX formed bigger foci (e.g., clusters) in LS-BZ-1 cells compared to LS-GD-1 cells. In addition, the analysis of telomeric dysfunction-induced foci (TIFs)—a hallmark of telomeric DNA damage [38]—assessed by immunofluorescence analysis of the co-localization between the fluorescence signal of γ-H2AX and that of the telomeric marker TRF2 revealed that the co-localization between the two markers did not markedly change in RHPS4-treated compared to untreated LS-GD-1 cells (Appendix A; Pearson’s correlation coefficient *r* = 0.54 ± 0.08 vs. 0.74 ± 0.08; Manders’ correlation coefficient M_2_ = 0.24 ± 0.04 vs. 0.16 ± 0.05 in treated and untreated cells, respectively) [32]. On the other hand, the extent of the overlap between the fluorescence signal of γ-H2AX and that of TRF2 (Pearson’s correlation coefficient *r* = 0.66 ±0.03 vs. 0.38 ± 0.05; Manders’ correlation coefficient M_2_ = 0.37 ± 0.02 vs. 0.08 ± 0.03 in treated and untreated cells, respectively) indicated that RHPS4-induced DNA damage localized, at least in part, at telomeric levels in LS-BZ-1 cells (Appendix A). Nonetheless, the exposure of DDLPS cells to an equitoxic concentration of RHPS4 triggered a pronounced accumulation of micronuclei (Appendix A and Figure 3C), a possible hint of dysfunctional telomeres, a significant fraction of which exhibited extensive DNA damage, as evidenced by the marked staining for γ-H2AX (Figure 3C and Appendix A). In addition, remarkable perturbations in the levels of telomeric c-circles were appreciable in RHPS4-treated cells. In particular, despite some fluctuations in the basal levels of c-circles observed in untreated DDLPS cells (Figure 3D), the exposure of LS-GD-1 cells to RHPS4 resulted in a time-dependent increased production of c-circles (Figure 3D), whereas a complete abrogation over time of c-circle levels was appreciable in LS-BZ-1 cells exposed to an equitoxic concentration of the compound (Figure 3D).

The time-dependent assessment of cell growth of DDLPS cells exposed to an equitoxic concentration of RHPS4 showed that the compound remarkably impaired the growth of both cell lines, as revealed by the number of growing cells that remained nearly constant over time in treated compared to untreated cells (Figure 4A). This effect was accompanied by a remarkable reduction of the in vitro migrating capabilities of LS-GD-1 cells. In particular, a significant (*p* < 0.001) reduction in the wound-healing capabilities of LS-GD-1 cells during a scratch assay (Figure 4B) was appreciable after 24-h exposure to RHPS4 (32% ± 2.9% of wound closure vs. T_0_) compared to untreated cells (65% ± 2.5% of wound closure vs. T_0_) (Figure 4B,C). Similarly, a significant reduction in the number of migrating LS-GD-1 cells (−60% ± 5.2% in treated vs. untreated cells; *p* < 0.05) was observed in a transwell assay upon 24-h exposure to RHPS4 (Figure 4C). Unfortunately, it was not possible to obtain reliable data on LS-BZ-1 cells due to their very poor in vitro migrating capabilities, making them unsuitable to be tested in the wound healing and transwell assays.

In addition, remarkably increased amounts of cleaved PARP-1—generated during apoptosis by activated caspases—were appreciable in LS-BZ-1 cells in response to 72-h exposure to RHPS4 (Figure 4D). This observation is in line with the evident accumulation of γ-H2AX clusters (Appendix A) and gained support by fluorescence microscopy analyses showing that 72-h exposure to RHPS4 resulted in a remarkable increase in the percentage of LS-BZ-1 cells (9.3 ± 0.75% vs. 1.0 ± 0.27% of the overall untreated cell population; *p* < 0.05) exhibiting an apoptotic nuclear morphology (Figure 4E). Conversely, no drug-induced apoptosis, in terms of cleaved PARP-1, was appreciable in the residual LS-GD-1 cell population upon 72-h exposure to an equitoxic concentration of RHPS4 (Figure 4D). Based on this evidence and the notion that micronuclei accumulation is often associated with the induction of autophagy, immunofluorescence analysis of the autophagy-associated marker LC3B (microtubule-associated protein 1 light chain 3 beta [39]) was performed in both untreated and RHPS4-treated DDLPS cells. Specifically, a diffuse LC3B-associated fluorescence signal was observed in both untreated DDLPS cell lines (Figure 4F). Conversely, a typical autophagosome-associated LC3B punctuate fluorescence pattern, suggestive of autophagy occurrence [39], was readily appreciable in the residual LS-GD-1 cell population, but not in LS-BZ-1 cells, after 72-h exposure to RHPS4 (Figure 4F). This finding gained further support with the evidence that the amounts of the LC3B-II protein—the lapidated, autophagosome-associated form of LC3B [39]—as well as of the lysosome/autophagosome marker LAMP-1 [40] remarkably increased 72 h after the exposure of LS-GD-1 cells to RHPS4 (Figure 4D). Consistently with the immunofluorescence data, no accumulation of LC3B-II and of LAMP-1 protein amounts was appreciable in RHPS4-treated LS-BZ-1 cells (Figure 4D).

## 4. Discussion

The unsatisfactory outcome of patients with DDLPS and the low response rate of these tumors to currently available systemic chemotherapies emphasize the need for novel actionable targets and new therapeutic approaches. In this study, through a comparative analysis of gene expression profiles of paired WD and DD components from clinical DDLPS specimens, we identified pathways related to “telomere maintenance” as up-regulated in the DD compared to the WD component. To ascertain the potential of telomere as a novel therapeutic target for DDLPS, we exploited a G4 stabilizing-based approach using the acridine derivative RHPS4, a molecule showing high selectivity for telomeric G4 structures [26]. The ligand has been widely documented to cause perturbations of telomere architecture (i.e., telomere uncapping/telomere dysfunctions) [22] and to exert good anticancer activity, both as a single agent and in combination with conventional anticancer agents and radiation, in models of different human tumor types, other than soft-tissue sarcomas [22]. Here, we report, for the first time, the effect induced by RHPS4 in two patient-derived DDLPS cell lines that represent unique in vitro models of this rare, nevertheless aggressive, tumor type. 

Our data showed that RHPS4 exerted remarkable cytotoxic activity in both DDLPS cell lines, with IC_50_ values within the low micromolar range. In particular, DDLPS cells exposed to an equitoxic concentration of RHPS4 were characterized by a marked accumulation of micronuclei, which exhibited extensive DNA damage in a significant proportion, as suggested by the positive staining for the phosphorylated form on Ser139 of histone H2AX [41]. Notably, this evidence corroborates the notion that micronuclei may form in the presence of dysfunctional telomeres [42], as well as recent findings showing that distinct G4 ligands may trigger the formation of micronuclei in different human tumor cells [43,44,45]. 

Although similar impairment of cell growth over time and a comparable amount of micronuclei/cell were observed in both RHPS4-treated DDLPS cells, the biological outcome was different. In particular, LS-GD-1 cells exposed to RHPS4 for 72 h showed a level of DNA damage that was comparable to that of untreated cells, as well as a trend towards an increase in c-circle production. Conversely, LS-BZ-1 cells exposed to an equitoxic concentration of RHPS4 showed an almost complete abrogation of c-circle production with respect to untreated cells, which was paralleled by the prominent accumulation of DNA damage. In particular, LS-BZ-1 cells treated with the G4 ligand were characterized by the accumulation of γ-H2AX foci in the form of clusters, thus suggesting that the DNA damage occurring in LS-BZ-1 cells could be more refractory to be repaired compared to that of LS-GD-1 cells [46].

The remarkable differences in the pattern of TRFs observed in the two DDLPS cells would account for the superior susceptibility of LS-BZ-1 cells, even to small perturbations induced by the compound to the telomere architecture. Moreover, different efficiency in recruiting telomere protecting factors, such as TRF2 or other shelterin components, could also account for the diverse susceptibility to DNA damage induction and occurrence of TIFs in LS-GD-1 compared to LS-BZ-1 cells in response to RHPS4. In this regard, it has been documented that C-rich telomeric variant repeats alongside non-telomeric SV40 DNA are dispersed throughout long (ALT) telomeres [47]. These non-conventional sequences may have remarkable implications for the structure/function of telomeric nucleoprotein and may favor a recombinogenic telomeric state, which sustains ALT activity ([47]). Whether telomeric variants are present in our DDLPS cells and whether they may impinge or not on the biological effects exerted by telomeric G4 ligands remain to be established. In addition, we found that the two DDLPS cell lines express markers of ALT and are characterized by telomerase activity. This observation supports previous evidence that multiple TMMs may be activated in LPS [12,24]. Though it remains unclear whether both TMMs are active in individual cells or whether the bulk cultures contain cell populations that use a single TMM [12], this evidence implies that the prevalence of a TMM with respect to the other may provide the bulk culture with distinct, or even opposite, biological behaviors [12], including the response to G4 ligand exposure.

Furthermore, RHPS4-mediated DNA damage induction and micronuclei accumulation resulted in an apoptotic response in LS-BZ-1 cells, as revealed by the assessment of nuclear morphology as well as by the accumulation of the cleaved form of PARP-1, while LS-GD-1 cells exposed to an equitoxic concentration of the compound failed to undergo apoptosis. Conversely, they were characterized by autophagic features, as indicated by the occurrence of the typical autophagosome-associated LC3B punctuate fluorescence pattern as well as the accumulation of both the lipidated, autophagosome-associated form of LC3B [39] and the lysosome/autophagosome marker LAMP-1 [40]. 

Autophagy is an evolutionarily conserved cellular process by which cellular waste is trafficked from the cytoplasm to lysosomes via membrane compartments (i.e., autophagosomes) for breakdown and eventual recycling [48]. In this regard, it has been demonstrated that the exposure of cancer cells to a telomeric G4 ligand derived from the anthracene resulted in the appearance of biochemical and morphological features typically associated with autophagy [22]. In particular, it has been documented that G4-ligand-induced autophagy occurred as a consequence of DNA damage induction and telomere uncapping and acted as a safeguard mechanism to counteract telomeric G4 ligand-mediated cellular stress [22]. Notably, our findings show that while RHPS4-mediated apoptosis induction in LS-BZ-1 cells was associated with the complete inhibition of c-circle production, which is consistent with impaired ALT activity, the exposure of LS-GD-1 cells to an equitoxic concentration of the compound resulted in a mild and time-dependent increase in the production of c-circle DNA, which may be suggestive of a proficient ALT activity, associated with the occurrence of autophagy-related features. This evidence would suggest that RHPS4-treated LS-GD-1 cells are able to keep ALT activity on course, hence cope with G4 ligand-induced cell stress—as also indicated by the lack of overt DNA damage—likely by activating cytoprotective autophagy [22]. Overall, our evidence is in keeping with the notion that autophagy exerts a primary cytoprotective role in response to different stimuli, including anticancer therapies [48] as well as with recent findings showing that RHPS4-induced telomeric damage fuels the overactivation of telomeric recombination (i.e., increased ALT activity) in osteosarcoma cells [49]. Moreover, it has been demonstrated that the activation of autophagy may rely on the accumulation of micronuclei, which, in turn, may be subjected to autophagic degradation [42,50]. Micronuclei are a source of cytoplasmic DNA that can be recognized by the cyclic GMP–AMP synthase (cGAS)-stimulator of interferon genes (STING) DNA sensor, leading to the activation of innate immunity [42]. Notably, cGAS has been also found to localize at DNA bridges, which represent markers of telomere dysfunctions, thus raising the possibility that G4 binder-mediated dysfunctional telomeres may also play a role in innate immune signaling [42]. Noteworthily, enhanced micronuclei formation in breast cancer cells exposed to G4 binders has been recently reported to trigger the activation of innate immune genes (i.e., type I interferon genes) via cGAS-STING [44].

Although we did not investigate this aspect in the present study, this evidence leaves enough space for future research directions. In this context, DDLPS is considered a sarcoma histotype suitable for immune therapy approaches, as reported in the SARC-028 clinical trial, where 20% of patients with metastatic DDLPS showed tumor responses to pembrolizumb (i.e., anti-PD-L1 antibody) [51]. Strategies that enhance tumor immune infiltrate before treatment with immune checkpoint inhibitors may not only improve the response rate in tumors harboring an immune infiltrate but also increase immune infiltrate in non-responsive tumors, as shown in other solid neoplasms [52,53]. Hence, by favoring the accumulation of cytoplasmic DNA fragments directly or indirectly via oxidative stress [54], G4 binders may exert an immunomodulating activity that could be harnessed for an immunotherapy-based approach [44,54]. 

## 5. Conclusions

The present study indicates that the G4 ligand RHPS4 remarkably inhibits the growth of two MDM2 amplified DDLPS patient-derived cell lines. The assessment of biological activity revealed that the exposure of DDLPS cells to an equitoxic concentration of RHPS4 induced telomere-related perturbations associated with the appearance of an apoptotic or autophagic phenotype in a cell type-dependent manner. This evidence is in keeping with the notion that the cell’s genetic background may steer the biological effects expected to arise from ligand-mediated G4 targeting in a cell context-dependent manner [22]. In this regard, it has been documented that TMPyP4, a cationic porphyrin acting as a telomeric G4 ligand and showing telomerase inhibitory activity, was able to stimulate lung cancer and osteosarcoma cell invasion, hence tumor progression, when administered at low concentrations (<0.5 μM), owing to its ability to interfere with the expression levels of genes functionally related to cell adhesion [55]. In addition, we cannot exclude that additional molecular factors, other than telomeres, which are relevant for the growth or survival of DDLPS cell lines (e.g., MDM2; CDK4, HMGA2 [27]) could be differently amenable to RHPS4-mediated G4 stabilization, thus accounting for the differences observed in terms of biological responses in our DDLPS models. Indeed, several G4 ligands belonging to distinct chemical families have been reported to show “promiscuous” activities based on their binding to multiple G4 elements within the human genome [22]. In addition, cell factors (e.g., specific G4 interacting proteins) or cell conditions (e.g., chromatin status or transcriptional activity) may impinge on the ligand–G4 target interaction and, consequently, on the biological outcomes expected to arise from ligand-mediated G4 targeting in a cell type-dependent manner [22]. 

Overall, our findings support G4 ligands as intriguing therapeutic tools in DDLPS and highlight that targeting G4 structures may represent an innovative and fascinating approach for the disease that is worth a deeper investigation. In this regard, it is worth noting that two G4 ligands, namely Quarfloxin and Pidnarulex, are currently being tested in clinical trials for different malignancies (www.clinicaltrial.gov, accessed on 16 May 2022).

## Figures and Tables

**Figure 1 cancers-14-02624-f001:**
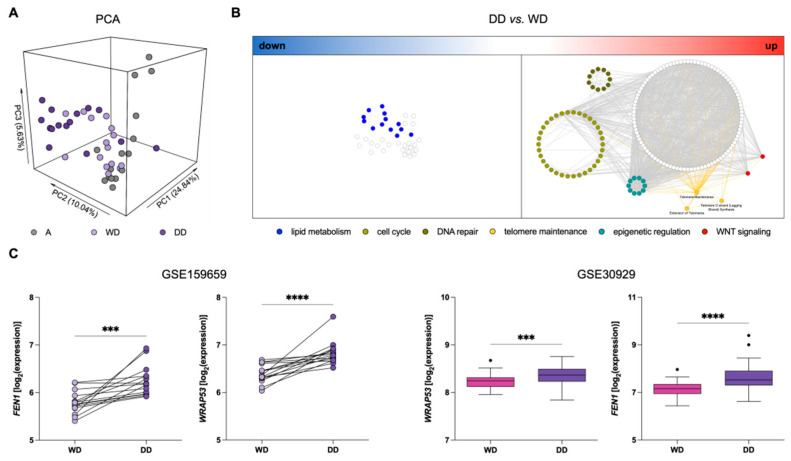
Gene expression profiles in DDLPS clinical samples. (**A**) Principal component analysis (PCA) showing segregation among different types of samples: Normal adipose tissue (A; grey), well-differentiated (WD; light purple) and dedifferentiated (DD; dark purple) components of DDLPS. (**B**) Network of over-represented pathways in DD vs. WD components of DDLPS. A big cluster represented by “lipid metabolism” (blue nodes) is among the down-regulated pathways (**left**), whereas “cell-cycle” (light green nodes), “DNA repair” (dark green nodes), “epigenetic regulation” (aquamarine nodes), “WNT signaling” (red nodes), and “telomere maintenance” (yellow nodes and edges) are highlighted among the significantly up-regulated pathways (**right**). (**C**) Dotplot (**left**) and Tukey’s boxplots (**right**) of WRAP53 and FEN1 expression levels in the different tissue components: Wilcoxon test for GSE159659 paired samples (**left**) and Mann–Whitney test for GSE30929 (**right**) samples. *** *p* < 0.001, **** *p* < 0.0001.

**Figure 2 cancers-14-02624-f002:**
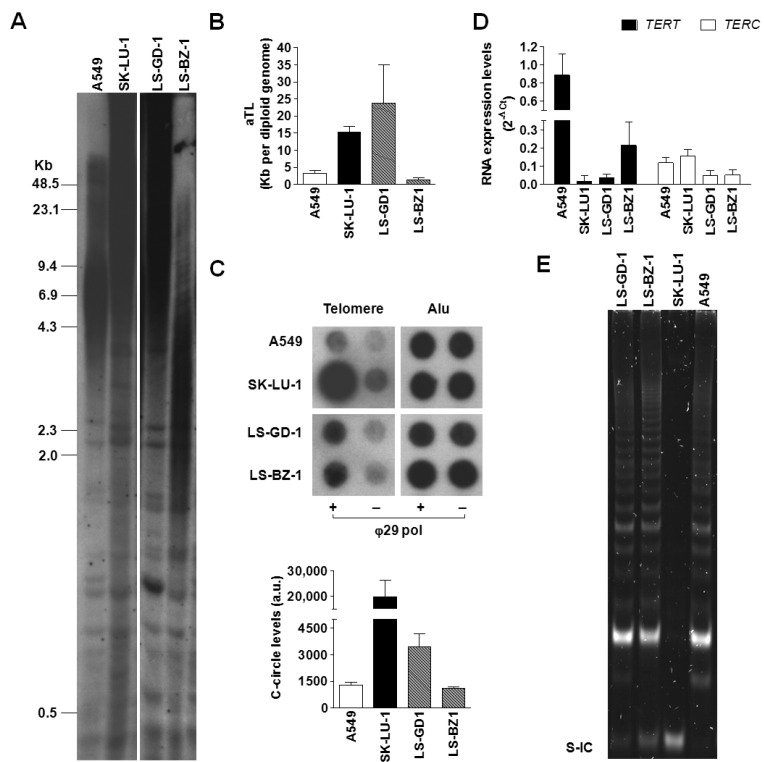
Characterization of TMM in DDLPS cell lines. (**A**) Representative image showing the distribution of TRFs assessed by Southern blot on total DNA obtained from LS-GD-1 and LS-BZ-1 cells. Numbers on the left indicate the molecular size (Kb, kilobases) of DNA fragments; (**B**) Real-time PCR quantification of absolute telomere length (aTL). Data have been reported as Kb per diploid genome, according to the manufacturer’s instructions, and represent mean values ± s.d. from at least three independent experiments. (**C**) Representative image of a dot blot from a CCA showing the basal levels of telomeric C-circles in total DNA from LS-GD-1 and LS-BZ-1 cells. The graph on the bottom shows the quantification of c-circle levels. Data have been reported as means ± s.d. of [(ϕ29+) − (ϕ29−)] densitometric values of telomeric probe from three independent experiments; a.u.: Arbitrary units. (**D**) Real-time RT-PCR assessment of the expression levels of human telomerase reverse transcriptase subunit (TERT) mRNA and its long non-coding RNA partner (TERC). Data have been reported as 2^−ΔCt^ and represent mean values ± s.d. (**E**) Representative image of a TRAP assay carried out on protein extracts obtained from LS-GD-1 and LS-BZ-1 cells. S-IC: 36 bp internal standard. Total DNA/RNA and protein extracts from the telomerase-positive A549 and SK-LU-1 cell lines have been included as internal controls in each assay. For the original, uncropped blots/gels see Appendix A.

**Figure 3 cancers-14-02624-f003:**
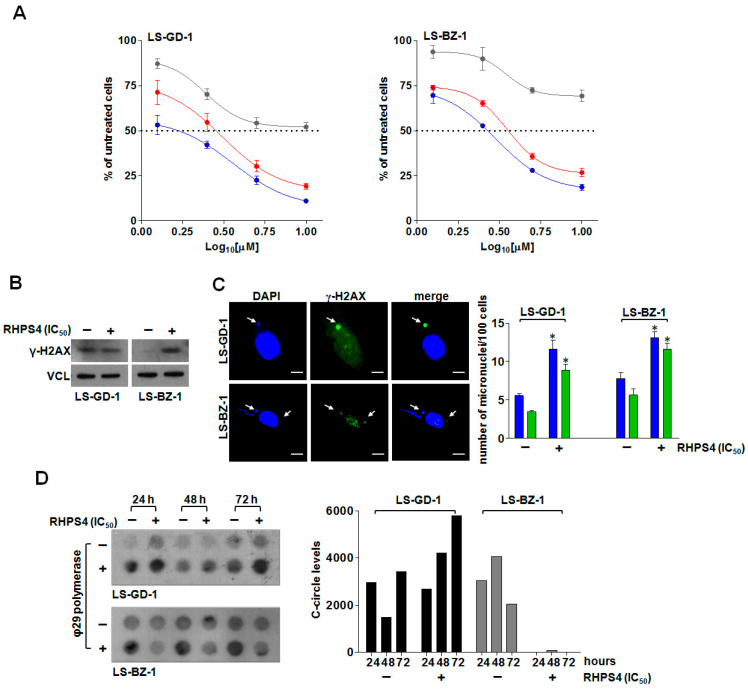
Cytotoxic activity and telomere-related effects of RHPS4 on DDLPS cells. (**A**) Dose–response curves for LS-GD-1 (left) and LS-BZ-1 (right) cells exposed for 24 (grey), 48 (red), and 72 (blue) hours to increasing concentrations of RHPS4. Data have been reported as percentage of growing cells with respect to untreated cells as a function of the Log10 of compound concentrations and represent mean values ± s.d. from at least three independent experiments. Dotted lines highlight 50% of cell growth inhibition. (**B**) Representative Western immunoblotting showing γ-H2AX protein amounts in LS-GD-1 and LS-BZ-1 cells exposed for 72 h to solvent (−) or to an equitoxic (IC_50_) amount of RHPS4 (+). Vinculin (VCL) was used as loading control. Cropped images of selected proteins are shown. (**C**) Representative image of micronuclei (red arrows) in DDLPS cells exposed for 72 h to an equitoxic (IC_50_) amount of RHPS4. Magnification: ×100, scale bar: 10 μm. The graph on the right shows the fraction of untreated (−) or RHPS4-treated (+) cells within the overall cell population that scored positive for micronuclei (blue bars) and for γ-H2AX-stained micronuclei (green bars). Data have been reported as number of micronuclei/100 cells and represent mean values ± s.d. from at least three independent experiments. * *p* < 0.05. (**D**) Representative image of a dot blot showing the time-dependent assessment of c-circle levels in LS-GD-1 and LS-BZ-1 cells exposed to solvent (−) or to an equitoxic (IC_50_) amount of RHPS4 (+). The graph on the right shows the quantification of c-circle levels. Data are reported as means of [(ϕ29+) − (ϕ29−)] densitometric values of telomeric probe from two independent experiments. An Alu probe was used to ensure equal sample loading (Appendix A). For the original, uncropped blots/gels see Appendix A.

**Figure 4 cancers-14-02624-f004:**
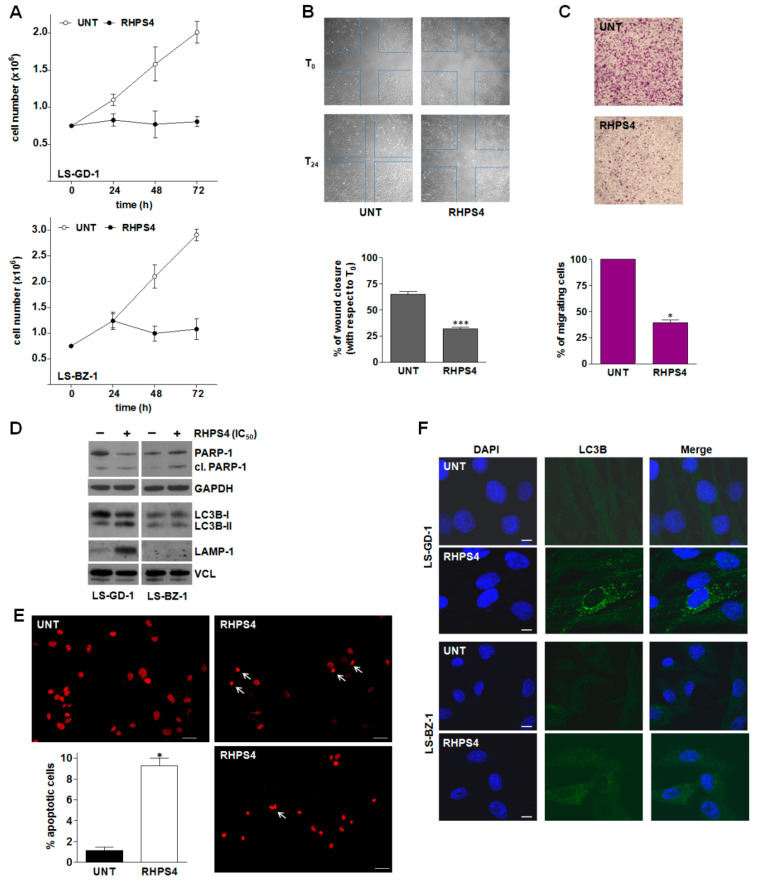
RHPS4-mediated biological effects on DDLPS cells. (**A**) Growth kinetics of untreated (ο) and RHPS4-treated (●) LS-GD-1 (top) and LS-BZ-1 (bottom) cells. Data have been reported as the number of growing cells and represent mean values ± s.d. from at least three independent experiments. (**B**) Representative images of a wound-healing assay showing the migration of untreated and RHPS4-treated LS-GD-1 cells at T_0_ and T_24_ (see material and methods); magnification ×4. The quantification of wound closure after 24-h exposure to RHPS4 (IC_50_) has been reported in the graph on the bottom. Data have been reported as the percentage of wound closure at T_24_ vs. T_0_ and represent mean values ± s.d. from at least three independent measurements; *** *p* < 0.001). (**C**) Representative image of cell migration assessed at 24 h by a transwell assay in untreated and RHPS4-treated LS-GD-1 cells, magnification ×4. The graph on the bottom reports the quantification of migrating cells. Data have been reported as percentage of migrating cells in RHPS4-treated vs. untreated cells and represent mean values ± s.d. from at least three independent experiments; * *p* < 0.05). (**D**) Representative Western immunoblotting showing the amounts of the indicated proteins in untreated (−) DDLPS cells and after 72-h exposure to an equitoxic concentration of RHPS4 (+). GAPDH and Vinculin (VCL) were used to ensure equal protein loading. Cropped images of selected proteins are shown. For the original, uncropped blots/gels see Appendix A. (**E**) Representative images of untreated and RHPS4-treated LS-BZ-1 cells stained with propidium iodide showing the apoptotic nuclear morphology at 72 h; scale bars: 50 µm; magnification: ×20. The graph shows the quantification of apoptotic cells in the overall LS-BZ-1 cell population. Data have been reported as the percentage of apoptotic cells and represent mean values ± s.d. from at least three independent counts. * *p* < 0.05. (**F**) Representative images of fluorescence microscopy analysis of LC3B in untreated DDLPS cells and after 72-h exposure to RHPS4. Nuclei were counterstained with DAPI. Merged images are shown; scale bar: 10 μm.

**Table 1 cancers-14-02624-t001:** In vitro cytotoxic activity of RHPS4 on DDLPS cell lines, reported as IC_50_ (μM) at 48 and 72 h.

Cell Line	48 h	72 h
LS-GD-1	3.2 ± 0.3	1.7 ± 0.1
LS-BZ-1	4.1 ± 0.2	2.5 ± 0.1

## Data Availability

The dataset and materials used during the current study are available from the corresponding authors on reasonable request.

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
