# Peer review of "Telomere as a Therapeutic Target in Dedifferentiated Liposarcoma"

_cancers, 2022, doi:10.3390/cancers14112624_

Round 1

Reviewer 1 Report

DDLPS is type of high grade liposarcomas with aggressive behavior, but lack of proper therapeutic. It is urgent to develop potential approaches to cure the disease. The authors analyzed the gene expression profile of paired samples from DDLPS patients, and found the upregulation of TMM pathways in DD type of DDLPS. Then they found that the G4 stabilizer, RHPS4 reduced cell proliferation and migration in MDM2 amplified DDLPS patient-derived cell lines. These data suggested that targeting TMM might be a prominent therapeutic approach in DDLPS. The authors must strengthen the arguments before publication.

In Figure 1, the authors stated that Telomere maintenance is highlighted as a significantly upregulated pathway. However, it only takes a very small portion of the upregulated proteins. Besides, in Figure 1C, the authors didn’t present the representative TMM proteins. The authors should better describe why they want to target TMM, since now there seems to be no effective cancer therapeutic approach is targeting TMM.

Figure 2B and Figure S5 (raw data of Fig 2B) (Table 1): The authors should note that LS-GD-1 and LS-BZ-1 CCA blot is using a high exposition compared to A549 and SK-LU-1 with low exposition. It indicates the so-called C-circle level could be extremely low. Besides, Table 1 can be included in Figure 2.  

Figure 2C & 2D. The authors showed higher TERT expression in LS-BZ1, but lower telomerase activity compared to LS-GD-1. How to interpret these inconsistent results?

In Figure 3, the authors tried to use RHPS4 to target telomere. The authors should examine the telomere damage upon RHPS4 treatment. It looks like RHPS4 didn’t directly induce telomere defects.

In previous study, Zheng et al. reported another G4 stabilizer TMPyP4 stimulate cell migration at low doses, but induces cell death at high doses (Xiao-Hui Zheng et al. 2016, Scientific Reports). It is worth examining the dosage response upon treatment with RHPS4.

Figure S2. The image of “The hybridization of CCA dot blot with an Alu probe” is saturated, which cannot be used as a loading control. Please show an image with proper exposure/ adjustment.  

Figure S3. The DAPI staining is fuzzy, while micronuclei are not visualized properly. Better image is required to support the argument.

The authors can provide flow cytometry result using PI-Annexin V staining kit to support the data of RHPS-4 induces cell death in DDLPS cells. And the authors should emphasize how and why different DD type of DDLPS is more sensitive to RHPS-4.

Author Response

Manuscript ID: cancers-1715083

Reviewer #1

We thank the Reviewer for her/his kind evaluation of our manuscript and for the valuable comments she/he has provided. Enclosed please find point-by-point replies to the Reviewer’s comments.

Comments and Suggestions for Authors

DDLPS is type of high grade liposarcomas with aggressive behavior, but lack of proper therapeutic. It is urgent to develop potential approaches to cure the disease. The authors analyzed the gene expression profile of paired samples from DDLPS patients, and found the upregulation of TMM pathways in DD type of DDLPS. Then they found that the G4 stabilizer, RHPS4 reduced cell proliferation and migration in MDM2 amplified DDLPS patient-derived cell lines. These data suggested that targeting TMM might be a prominent therapeutic approach in DDLPS. The authors must strengthen the arguments before publication.

1) In Figure 1, the authors stated that Telomere maintenance is highlighted as a significantly upregulated pathway. However, it only takes a very small portion of the upregulated proteins. Besides, in Figure 1C, the authors didn’t present the representative TMM proteins. The authors should better describe why they want to target TMM, since now there seems to be no effective cancer therapeutic approach is targeting TMM.

We apologize with the Reviewer for the misunderstanding. In fact, Figure 1B shows the over-representation analysis of the significant differentially expressed (DE) genes (not proteins) in DD vs. WD comparison. The applied method determines whether genes from pre-defined sets (e.g., those belonging to a specific REACTOME pathway) are present more/less than would be expected (i.e., over-/under-represented) in a subset of own data. In addition, for each pre-defined set a statistical test is applied to calculate the significance in terms of adjusted p-value, to take into account of multiple test comparison, for the specific over-representation with respect to the fact that this is obtained by chance. In line with this evidence, even if TMM-related genes take a very small portion of the up-regulated genes, they remain statistically significant. Anyway, we have now provided a new version of Figure 1, also to accomplish with the request raised by Reviewer #2. We must also apologize with the Reviewer as during the drawing of the original Figure 1 the legend label attributed to TMM nodes was inverted with that of the epigenetic regulation nodes. However, this misleading information has now been amended in the new version of Figure 1, where the connection between TMM related gene sets and other up-regulated gene sets have been also highlighted. In addition, a novel Table S1 has been included in the revised version of the paper to disclose the annotation clusters of significantly over-represented REACTOME pathways in the DD vs. WD comparison, where gene symbols within each pathway have been also included. This additional information indicates the reason why we showed FEN1 and WRAP53 gene expression levels in the two independent data sets reported in Figure 1C, being the two genes the most closely related to telomere maintenance among those included in each up-regulated TMM-related pathway (page 7, lines 335-340 in the original version of the manucript).

Finally, in the main text of the original version of our manuscript we mentioned the reasons why we try to target TMM in DDLPS (Introduction, lines 85-138; Results, lines 333-335; Discussion, lines 535-552 in the original version of the manuscript). In particular,

  1. a) The activation of TMM is an almost overriding feature of human cancers and they have been regarded as intriguing cancer-associated targets (ref 12);
  2. b) It has been documented that TMMs are relevant for DDLPS progression, and, consequently, prognostic risk stratification ( 12, 22, 23). This evidence points to telomere as intriguing target for DDLPS.
  3. c) Since the evidence of the selective reactivation in most human tumors, telomerase has gained attention as a possible cancer-associated target and several strategies to interfere with its expression and functions for potential therapeutic applications have been widely pursued ( 19). In this context, it should be also taken into account that Imetelstat has been the first telomerase inhibitor to enter clinical trials in patients with solid tumors and myeloproliferative diseases (Relitti N, et al. Telomerase-based Cancer Therapeutics: A Review on their Clinical Trials. Curr Top Med Chem. 2020;20(6):433-457. doi: 10.2174/1568026620666200102104930);
  4. d) Telomeres have been considered as biologically relevant targets of small molecules able to interact and stabilize G-quadruplex (G4) structures (ref. 20), especially for tumors relying on ALT mechanisms for which no genuine ALT targeting therapies have been developed yet ( 12). In the original version of the manuscript we mentioned that “our findings support G4 ligands as intriguing therapeutic tools in DDLPS and highlight that targeting G4 structures may represent an innovative and fascinating approach in the disease that worth of deeper investigation” (Conclusions, lines 536-538 in the original version of the manuscript). In this context, Quarfloxin has been the first G-quadruplex ligand to enter clinical trials for patients with solid tumors, lymphoma and neuroendocrine carcinoma (ClinicalTrials.gov Identifier: NCT00780663; NCT00955292; NCT00955786). Novel G4 ligands are now being tested in clinical trials, including Pidnarulex (ClinicalTrials.gov Identifier: NCT04890613) and QN-302 (Qualigen).

2) Figure 2B and Figure S5 (raw data of Fig 2B) (Table 1): The authors should note that LS-GD-1 and LS-BZ-1 CCA blot is using a high exposition compared to A549 and SK-LU-1 with low exposition. It indicates the so-called C-circle level could be extremely low. Besides, Table 1 can be included in Figure 2.

We agree with the reviewer’s comment and apologize for the misleading information. Indeed, we showed in the original Figure 2B a lower exposure image for the CCA controls since SK-LU-1 cells have very high amounts of c-circles.

We agree that DDLPS cells are characterized by low levels of c-circle DNA, as we originally showed in Table 1 and mentioned in the original main text, pointing out that they are likely characterized by both TA and ALT. However, in the revised version of the paper we have now provided a novel Figure 2B showing the same exposure time for A549, SK-LU-1 and DDLPS cells. The level of c-circles in A549 has been also shown in the graph of panel 2C and text modified accordingly. In addition, to accomplish with the Reviewer request, Table 1 has been removed from the main text and data have been shown in the novel Figure 2B and C as graphs. Please note that to fulfill the request raised by Reviewer #2 the values for aTL have now been expressed as Kb instead of base pairs.

3) Figure 2C & 2D. The authors showed higher TERT expression in LS-BZ1, but lower telomerase activity compared to LS-GD-1. How to interpret these inconsistent results?

We apologize for this misunderstanding. In our opinion, there is no biological reason to expect that the levels of a given mRNA should correlate with the catalytic activity of the coded enzyme. In addition, it should be taken into account that i) the TaqMan probe we used to assess TERT expression levels recognizes a sequence located at the boundary between Ex3 and Ex4 and the amplicon obtained upon amplification is common to all TERT splicing variants described thus far (see for instance Biomedicine 2021, 9, 526). Consequently, we cannot excluded that a different combination of TERT splicing products - considering for instance that the alpha-variant of TERT has been reported to act as a dominant negative - would account for the supposed lower telomerase activity in LS-BZ-1 cells compared to LS-GD-1 cells; ii) it is widely recognized that TERT undergoes several different posttranslational modifications, a fine balance of which may dictate the activation/inactivation of TERT catalytic activity. A possible difference in these TERT post-translation modifications would account for the supposed difference in the levels of telomerase activity between LS-BZ-1 and LS-GD-1 cells; iii) we did not intend to perform telomerase activity quantification. The image we showed in the original version of our manuscript was intended for qualitative purposes (presence or absence of telomerase activity). Indeed, A549 cells showed the highest levels of TERT mRNA but a level of telomerase activity comparable to that, for example, of LS-GD-1 cells (Figure 2).

4) In Figure 3, the authors tried to use RHPS4 to target telomere. The authors should examine the telomere damage upon RHPS4 treatment. It looks like RHPS4 didn’t directly induce telomere defects.

We understand the concern raised by the reviewer. Telomeric-induced DNA damage was originally reported in Figure S1. Our data in Figure 3B show that untreated LS-GD-1 cells are characterized by higher basal levels of DNA damage with respect ot LS-BZ-1 cells,  which has been reported to be a common feature associated with ALT activity (Lovejoy CA, et al. PLoS Genet. 2012;8(7):e1002772), together with modestly elevated mitochondrial dysfunction and reactive oxygen species (ROS) production (Hu J, et al. Cell. 2012 Feb 17;148(4):651-63). Unexpectedly, a 72-h exposure to RHPS4 did not result in a dramatic increase in g-H2AX protein levels (Figure 3B), as we have originally described in the main text.

However, to accomplish the Reviewer’s request we included in the revised version of the manuscript a novel Figure S1 showing immunofluorescence of g-H2AX and TRF2, where previous images alongside novel panels have been assembled and properly adjusted. This additional data support the evidence that LS-GD-1 cells has high basal level of DNA damage as well as that there is no such a remarkable increase in it upon a 72-h exposure to RHPS4. In addition, zoomed images in the novel figure S1 show that the fraction of DNA damage foci that co-localize with TRF2 is almost comparable in treated and untreated LS-GD-1 cells. Conversely, LS-BZ-1 cells are characterized by overt DNA damage (Figure 3B and novel Figure S1), that at least in part co-localizes at telomeric level, upon exposure to an equitoxic concentration of RHPS4. Notably, DNA damage assessed by immunostaining indicates that g-H2AX formed bigger foci (e.g., clusters) in LS-BZ-1 cells compared to LS-GD-1 cells, thus suggesting that this damage could be more refractory to be repaired (Asaithamby A et al. PNAS 2011; 108:8293) compared to that of LS-GD-1 cells. In this regard, the main text has been modified accordingly.

In addition, in Figure 3C we showed in both RHPS4-treated DDLPS cell lines a significantly increase in the number of unstained and g-H2AX-stained micronuclei, which have been reported to likely form in the presence of dysfunctional telomeres (page 11, lines 458-461; page 13, lines 558-560 in the original version of the manuscript).

5) In previous study, Zheng et al. reported another G4 stabilizer TMPyP4 stimulate cell migration at low doses, but induces cell death at high doses (Xiao-Hui Zheng et al. 2016, Scientific Reports). It is worth examining the dosage response upon treatment with RHPS4.

We thank the Reviewer for this suggestion, though we have to point out that we intentionally used an equitoxic concentration (IC50) of the compound to be more confident that the observed biological effects in the two DDLPS cell lines were directly comparable and mainly attributable to G4 stabilizing properties of the compound, rather than being associated to its general cytotoxic activity (dose-response curves reported in Figure 3A). In addition, such a working concentration of the compound allowed us to collect enough material for subsequent analyses, considering the both cell lines, which represent unique models of patent-derive DDLPS cells, are basically difficult to be maintained in culture.

In addition, we are a bit uncertain about the feasibility of a direct comparison of the effects exerted by two different compounds on different experimental models and under distinct experimental conditions. It should be also taken into account that a major hurdle for TMPyP4 deals with its ability to bind to duplex and triplex DNA, a characteristics that impinges on its selectivity for the binding to G4 structures (Sanchez-Martin V. Cancers 2021, 13, 3156).

However, the findings from the paper by Zheng et al. (now quoted as ref 55) have been discussed in the revised version of the manuscript.

6) Figure S2. The image of “The hybridization of CCA dot blot with an Alu probe” is saturated, which cannot be used as a loading control. Please show an image with proper exposure/ adjustment.

We apologize for the poor quality of this image. According to the Reviewer’s request, in the revised version of the manuscript figure S2 has been replaced with a panel showing a proper exposition/adjustment.

7) Figure S3. The DAPI staining is fuzzy, while micronuclei are not visualized properly. Better image is required to support the argument.

We apologize again for the poor quality of the images. According to the Reviewer’s comment, we included a new version of Figure S3 where the DAPI signal has been enhanced to make more appreciable the presence of micronuclei (Tang Z. et al., BMC Cancer, 2018; 18:426). Similarly, Figure 3C originally showing the presence of micronuclei has been adjusted accordingly.

8) The authors can provide flow cytometry result using PI-Annexin V staining kit to support the data of RHPS-4 induces cell death in DDLPS cells. And the authors should emphasize how and why different DD type of DDLPS is more sensitive to RHPS-4.

We thanks the Reviewer form this kind suggestion. However, it should be taken into account that we mentioned to have found hints of apoptosis only in LS-BZ-1 cells treated for 72 h with RHPS4 (IC50). In particular we have shown the occurrence of PARP-1 cleavage (Figure 4D) and the presence of nuclei with an apoptotic nuclear morphology (Figure 4E), that accounted for only 9% of the overall cell population (Figure 4E). On the basis of this premise we believe that PI-Annexin V staining would not be particularly relevant to obtain additional insights on apoptosis induction under our experimental conditions. In addition, we have to highlight that the assessment of PI-AnnexinV staining by flow cytometry could not be as much informative as expected since the fluorescence emission of RHPS4 would interfere with the emission of FITC-labeled AnnexinV, currently available in our lab. In fact, when we tried to dig on apoptosis on RHPS4-treated LS-BZ-1 cells by performing TUNEL (FITC labeling) assay, a remarkable shift in the staining of the overall treated cell population was observed, thus making these data non correctly interpretable. An example of this TUNEL assay performed on LS-BZ-1 cells according to our experimental conditions has been herein included for the reviewer knowledge. However, to provide more support to our evidence of apoptosis induction, we below include for the Reviewer knowledge a western blot showing a marked reduction in the amounts of some anti-apoptotic factors (such as Bcl-2, Claspin and Heme Oxygenase-1) in RHPS4-treated LS-BZ-1 cells that, alongside the cleavage of PARP-1, may indeed create a pro-apoptotic milieu.

The possible reasons for how and why DDLPS are differently sensitive to RHPS4 may be copious in number, and some of them, in our opinion, have been reasonably documented in the discussion and conclusion sections of the original manuscript (page 13, lines 558-561; page 14, lines 562-573 and lines 594-595 in original version of the manuscript).

Moreover, we have discussed on the evidence that G4-ligand induced autophagy may act as a safeguard mechanism to counteract G4 ligand-mediated cellular stress, thus acting as a “non conventional” resistance mechanism, as reported in ref. 21 (page 14, lines 576-582 in original version of the manuscript). In addition, we also reported that our findings show that while RHPS4-mediated apoptosis induction in LS-BZ-1 cells was associated to a complete inhibition of c-circle production (i.e., impaired ALT activity), the exposure of LS-GD-1 cells to an equitoxic concentration of the compound resulted in a mild and time-dependent increase in the production of c-circle DNA (i.e. proficient ALT activity) associated to the occurrence of autophagy-associated features. This evidence would suggest that RHPS4-treated LS-GD-1 cells were able to keep ALT activity on course, hence to cope with G4 ligand-induced cell stress likely by activating a cytoprotective autophagy (page 14, lines 582-589 in original version of the manuscript).

Finally, in the conclusions (page 15, lines 617-634 in the original version of the manuscript) we mentioned that cell genetic background may steer the biological effects expected to arise from ligand-mediated G4 targeting in a cell context-dependent manner. We also did not exclude that, other than telomeres, additional molecular factors which are relevant for the growth or survival of DDLPS cell lines could be differently amenable of RHPS4-mediated G4 stabilization, thus accounting for the differences observed in terms of biological responses. Indeed, several G4 ligands belonging to distinct chemical families have been reported to show “promiscuous” activities based on their binding to multiple G4 elements within human genome. In addition, cell factors (e.g., specific G4 interacting proteins) or cell conditions (e.g., chromatin status or transcriptional activity) may impinge on the ligand-G4 target interaction and, consequently, on the biological outcomes expected to arise from ligand-mediated G4 targeting in a cell type-dependent manner (page 15, lines 624-634 in original version of the manuscript).

Additional reasons, now discussed in the revised version of the manuscript, may also deals with the remarkable difference in telomere length observed in the comparison between LS-GD-1 cells, showing a pattern of TRFs consistent with an ALT phenotype (Figure 2), and LS-BZ-1-cells that instead are characterized by sort telomeres, even shorter than those detected in the telomerase-positive A549 cells, and consequently may be more susceptible to even little perturbations in telomere architecture induced by RHPS4. In addition, a different efficiency in recruiting telomere protecting factors, such as TRF2 or other shelterin components, in response to RHPS4 could also account for the diverse susceptibility to DNA damage induction and occurrence of TIFs in LS-GD-1 compared to LS-BZ-1 cells. In addition, it has been documented that C-rich telomeric variant repeats alongside non telomeric SV40 DNA are dispersed throughout long (ALT) telomeres, and may have remarkable implications for the structure and function of telomeric nucleoprotein (Conomos D. et al., Front Oncol. 2013;3:27). In this context, the spread of telomeric variants may destabilize the telomere in favor of a recombinogenic locked-in telomeric state, which creates a positive feedback loop that results in sustained ALT activity (Conomos D. et al., Front Oncol. 2013;3:27). This scenario would account, at least in part, for the observation that LS-GD-1 cells with typical ALT telomere length showed comparable level of DNA damage and a trend towards an increase in c-circle production after 72 h of RHPS4 exposure compared to untreated cells.

Finally, we found that the two DDLPS cell lines express markers of ALT and are characterized by telomerase activity. This observation supports previous evidence that multiple TMM may be activated in LPS. Though it remains unclear whether both TMMs are active in individual cells or the bulk cultures contains cell populations that use a single TMM [12], this evidence implies that the prevalence of a TMM with respect to the other may provide bulk culture with distinct, or even opposite, biological behaviors, including the response to G4 ligand exposure.

Reviewer 2 Report

The manuscript focuses on describing the effect of the G-quadruplex binder RHPS4 on patient-derived dedifferentiated liposarcoma cells. The manuscript is written well with logically build experiments and critically stated conclusions.

Please check the readability of the text in the figures before sending manuscripts for review. 

I suggest publishing the manuscript after addressing the following minor issues.

  1. Text in all figures and legends should be increased 2-fold.
  2. The description of data points within the Figure 1B part down is unreadable. Please consider redesigning part B down so the text from the figure is placed in the legend.
  3. Table 1 – Tabulated values should be rounded to two valid numbers. Consider using unit kilobase pairs (kbp). For example, 3340 ± 680 bp should be 3 ± 0.7 kbp.

Author Response

Manuscript ID: cancers-1715083

Reviewer #2

We thank the Reviewer for her/his very positive evaluation of our manuscript and for the accurate comments she/he has provided. Enclosed please find point-by-point replies to the Reviewer’s comments.

Comments and Suggestions for Authors

The manuscript focuses on describing the effect of the G-quadruplex binder RHPS4 on patient-derived dedifferentiated liposarcoma cells. The manuscript is written well with logically build experiments and critically stated conclusions.

Please check the readability of the text in the figures before sending manuscripts for review. 

I suggest publishing the manuscript after addressing the following minor issues.

1) Text in all figures and legends should be increased 2-fold.

All figures have been changed accordingly.

2) The description of data points within the Figure 1B part down is unreadable. Please consider redesigning part B down so the text from the figure is placed in the legend.

The whole Figure 1 has been redesigned, to fulfill with the request of Reviewer #1.

We must also apologize with the Reviewer as during the drawing of the original Figure 1 the legend label attributed to TMM nodes was inverted with that of the epigenetic regulation nodes. However, this misleading information has now been amended in the new version of Figure 1, where the connection between TMM related gene sets and other up-regulated gene sets have been also highlighted. In addition, a novel Table S1 disclosing the annotation clusters of significantly over-represented REACTOME pathways in the DD vs. WD comparison has been included to fulfill the request of Reviewer #1. We did not provide such kind of information for the under-represented pathways (i.e., lipid metabolism, blue points in Figure 1) since we did not dig on these biological circuitry in the present manuscript. However, we include herein a magnification of Figure 1B part down where the text of the labels is now readable, for the Reviewer’s own knowledge.

3) Table 1 – Tabulated values should be rounded to two valid numbers. Consider using unit kilobase pairs (kbp). For example, 3340 ± 680 bp should be 3 ± 0.7 kbp.

Thanks for the kind suggestion. The values have been modified accordingly to the Reviewer’s request. In particular, absolute telomere length has now been shown in a graph reported in panel B of the new figure 2 (also to accomplish with the request raised by Reviewer #1) and data expressed as Kb (Kilobases), rounded to two valid numbers. Similarly, the quantification of c-circle levels in DDLPS cells have now been shown as a graph in the novel figure 2 panel C.Table 1 has been deleted.

Round 2

Reviewer 1 Report

All my issues are addressed. Thanks